# OpenReview forum: "RAG-ENHANCED ASPECT-BASED SENTIMENT ANALYSIS FOR MOBILE APPLICATION REVIEWS: A MULTIAGENT FRAMEWORK FOR DEVELOPER-ORIENTED INSIGHT GENERATION"
_ICLR.cc/2026/Conference — Submitted to ICLR 2026_

### Official Review · Reviewer_hE4E · 2025-10-30

**Soundness:** 2
**Presentation:** 3
**Contribution:** 2
**Rating:** 4
**Confidence:** 4

**Summary:**

This paper proposes a framework for Aspect-Based Sentiment Analysis (ABSA) that integrates Retrieval-Augmented Generation (RAG) and a multi-agent system to provide mobile app developers with actionable insights and code-aware solutions from user reviews. The system is built upon a fine-tuned LLaMA 3.1 8B model with LoRA. Evaluated on a curated dataset of 41,245 app reviews, it reports high sentiment classification accuracy (98.23%) and aspect extraction F1 (82.26%). The authors claim superiority over several existing methods and emphasize the practical, production-ready nature of their system.

**Strengths:**

+ Rich System Design: The work describes an end-to-end framework that thoughtfully considers the entire pipeline from data ingestion and ABSA to solution generation.
+ Resource Efficiency: The use of LoRA and 4-bit quantization to reduce computational footprint is a positive practical contribution for leveraging open-source LLMs.
+ Valuable Problem Focus: The vision of bridging user feedback directly to code-level solutions is a highly relevant and important direction for the software engineering community.

**Weaknesses:**

- Lack of Novelty and Unclear Contribution: The core components of the proposed method (ABSA, RAG, multi-agent orchestration, LoRA) are widely-used and mature technologies. The paper fails to convincingly demonstrate a fundamental scientific contribution. It appears more as a competent engineering integration of existing tools rather than a novel methodological or theoretical advancement.

- Dataset Issues: The dataset source (45 apps) and composition are described vaguely, are not public, and cannot be verified.

- Technical details are ambiguous. The specific architecture of the "eight agents" in multi-agent systems, the communication protocols among them, and the decision-making processes are not described in depth enough. According to Figure 2, the workflow structure has been manually designed. However, the task of Coordinator is workflow orchestration, which leads to confusion.

**Questions:**

-	What is the most core scientific contribution of this paper that distinguishes it from the simple baseline integrated system?
-	How the multi agent work and collaborate?

---

### Official Review · Reviewer_oazG · 2025-10-31

**Soundness:** 2
**Presentation:** 2
**Contribution:** 2
**Rating:** 2
**Confidence:** 3

**Summary:**

This paper presents a well-engineered and highly practical multi-agent framework that connects Aspect-Based Sentiment Analysis (ABSA) of user reviews directly to source-code-aware solution generation. The system's primary strength is this novel and valuable integration, moving beyond mere analysis to provide actionable developer insights. The architecture, which uses LoRA-based SFT of LLaMA-3.1-8B orchestrated by LangGraph, is clearly described and achieves strong sentiment classification accuracy (98.23%). However, the paper suffers from several critical weaknesses. Methodologically, it is a complex integration of existing tools rather than a novel algorithm. The core contribution—code generation—is not rigorously evaluated; it lacks quantitative metrics for code quality and is not compared against any relevant code-generation baselines. Furthermore, the system is evaluated on a custom-built dataset, not a standard public benchmark, limiting claims of generalizability.

**Strengths:**

- **Novelty and Practicality of Code-Aware Solution Generation.** The paper's most significant strength is its novel "transforms raw user feedback into actionable developer insights", which directly addresses a key limitation in prior work that stops at analysis.
- **Comprehensive and Well-Designed System Architecture.** The system architecture is comprehensive and clearly illustrated, detailing the workflow from data ingestion to insight generation. The use of LangGraph to orchestrate eight specialized agents (e.g., Classifier, Context, Solutions) represents a sound and modular engineering design. This architecture effectively integrates contextual RAG using app changelogs and dual vector databases to flexibly handle different developer queries.

**Weaknesses:**

- **Methodological Integration without Algorithmic Novelty.** The paper presents a sophisticated and complex system but does not introduce new algorithms or fundamental methods. The entire framework is a well-engineered pipeline integrating existing, off-the-shelf components, including LoRA-based fine-tuning, LLaMA models, LangGraph orchestration, and standard RAG techniques.
- **Lack of Formalism and Mathematical Precision.** The methodology is described entirely in prose, lacking the mathematical formalism. Key components like the RAG retrieval function, the dual knowledge storage strategy, or the LoRA adaptation are not defined with any equations or formal notation, which reduces the technical clarity and precision of the proposed method.
- **Evaluation on a Non-Standard, Limited Dataset.** The system is evaluated on a custom-built dataset. While extensive, this dataset is not a recognized public benchmark, which makes it difficult to fairly compare the reported performance against other SOTA models.

**Questions:**

- Could you please elaborate on what you consider to be the core scientific or algorithmic contribution of this work, as distinct from the strong engineering integration?
- The methodology, particularly the RAG pipeline, is described entirely in prose, which can lack technical precision. To improve clarity and reproducibility, could you provide a more formal (perhaps mathematical) definition of your key components?
- Could you clarify why existing public benchmarks (e.g., SemEval for ABSA) were not used to provide a more direct, head-to-head comparison of the core sentiment and aspect extraction model?
- How was the claimed "24.6% improvement in solution implementation accuracy" measured, and what was the baseline?

---

### Official Review · Reviewer_9W2E · 2025-10-31

**Soundness:** 2
**Presentation:** 2
**Contribution:** 2
**Rating:** 2
**Confidence:** 4

**Summary:**

This paper presents a RAG-enhanced, multi-agent Aspect-Based Sentiment Analysis (ABSA) system for mobile app reviews. The system collects ~41k Google Play reviews, fine-tunes LLaMA-3.1-8B using LoRA, integrates dual-DB semantic retrieval (reviews + code), and orchestrates 8 LLM agents via LangGraph to extract aspects, sentiments, and opinions, then generate developer-actionable code fixes (Java/Kotlin/XML). The authors report strong performance (98.23% sentiment accuracy, ~82% aspect F1) and claim the system delivers production-ready developer feedback and code suggestions.

**Strengths:**

- Ambitious and practically relevant developer-tool pipeline.
- Solid data scale and domain focus (mobile apps).
- Effective engineering: LoRA, quantization, RAG, multi-agent routing.

**Weaknesses:**

1. **No standard ABSA evaluation → weak external validity**
2. **Self-curated dataset without annotation quality or release**
3. **Over-claimed novelty** relative to existing ABSA + RAG + code-repair research
4. **Lack of human study** or code correctness verification
5. **No ablation quantification** (only qualitative)
6. **Heavy system-engineering focus**, less scientific contribution
7. **Possible data leakage** (user reviews from public data → LLM training risk)

**Questions:**

1. Why not evaluate on SemEval & MAMS?
2. How was annotation done? How many annotators? Agreement metrics?
3. Is the dataset public? If not, how can others reproduce results?
4. How do you verify code suggestions are correct or actionable?
5. Did you compare to prompting GPT-4/Claude without multi-agents?
6. How much does each component contribute numerically (LoRA, dual DB, agents)?

---

### Meta-Review · Area_Chair_EuKT · 2025-12-18

**Summary:**

This paper leverages resource efficient fine-tuning of LLMs and retrieval augmented generation techniques to perform aspect-based sentiment analysis in review text data. Technical novelty and limited empirical evaluations are the shared concerns across all three reviewers. Moreover, the submission unfortunately disclosed its affiliation, which seriously violates the double-blind reviewing policy. As a result, we decided not to accept the submission.

**Reviewer Concerns:**

The authors did not provide any rebuttal.

**Reviewer Scores:**

There would not be any change.

---

### Decision · Program_Chairs · 2026-01-26

Reject